# Fatigue Analysis of Composite Bolted Joints under Random and Constant Amplitude Fatigue Loadings

**DOI:** 10.3390/ma17112740

**Published:** 2024-06-04

**Authors:** Song Zhou, Bo Yang, Yichang Huang, Xiaodi Wu

**Affiliations:** 1College of Civil Aviation, Northwest Polytechnical University, Taicang 215400, China; zhousong@nwpu.edu.cn; 2Shanghai Institute of Special Equipment Inspection and Technical Research, Shanghai 200063, Chinahuangyc@ssei.cn (Y.H.); 3School of Marine Engineering and Technology, Sun Yat-sen University, Zhuhai 519082, China

**Keywords:** composite, joint, constant amplitude fatigue, random fatigue, fatigue life

## Abstract

This paper attempts to analyze the random fatigue life and failure modes of joints using two calculation methods. Three kinds of tests were carried out, which were the static test, constant amplitude fatigue test and the random fatigue test, and four kinds of joints were designed. After the static test, the joint was subjected to a constant amplitude fatigue test by selecting different percentages of load according to the static strength. In order to predict the random fatigue life more precisely, two calculation methods were carried out, which were the linear cumulative damage method and the equivalent loading finite element method. Based on the linear cumulative damage hypothesis, the fatigue life of the joint was established as a function of the load amplitude, and then, the random life prediction was calculated by the amplitude distribution of the random loading. Another method was the equivalent loading method, which was to obtain the equivalent constant amplitude fatigue loading of the random loading spectrum. The finite element model was established based on the stiffness and strength degradation rule. The equivalent random life and fatigue failure modes of the joint were modeled. The two life prediction methods show good agreement with the fatigue experimental result, and all prediction results were included in a scatter band of the factor of 2.

## 1. Introduction

Composite bolted joints are widely adopted in aerospace, aviation and other engineering structures, and the joints are under complicated loading conditions [1,2,3,4], so their random fatigue properties are an important factor in structures.

Many researchers have investigated methods of joint fatigue life prediction, and the fatigue prediction models are divided into three categories: the fatigue life model, the residual stiffness and residual strength model and the cumulative damage model. Fatigue life prediction [5,6,7,8] is based on the S-N curve to characterize the fatigue properties of a composite, and the failure of the composite structure is judged by material failure criteria. This method requires a large amount of experimental data and does not focus on the actual damage mechanism in the fatigue process of composite materials, and the advantage of this method is that it can predict the fatigue life of a particular type of composite structure under specified conditions. The residual stiffness and strength model [9,10] uses a set of macroscopically measurable descriptive variables to analyze the damage of composite joints under fatigue loading. It mainly considers the degradation of material stiffness or strength during the fatigue process and that the critical load for structural failure decreases with the increase in the number of fatigue cycles. A lot of experiments are required to determine the parameters in the model. These models do not consider the actual damage mechanisms that occur during fatigue and belong to the empirical method of phenomenological theory. The cumulative damage model [11,12,13,14] uses one or more damage variables to describe the degree of internal damage and structural performance degradation during fatigue and quantitatively describes the actual damage accumulation during fatigue.

Many studies have been conducted on the fatigue life of composite laminates. However, research on the life of composite laminates with notched laminates is still very limited. It cannot be directly calculated using the fatigue life calculation method of non-porous composite laminates. The fatigue life prediction of composite joints is quite complicated. Several researchers have proposed their own fatigue prediction methods to predict the fatigue life of composite joints.

Schütz [15] proposed a fatigue strength–life fitting formula based on the fatigue strength obtained from the S-N curve in joint fatigue studies. This method is a semi-empirical formula, which can only predict the fatigue life of the joint and cannot determine the failure mechanism and failure mode of the joint. Iorio [16] proposed to use the overall stiffness and damping coefficient as the damage parameters and to determine the damping coefficient and elastic modulus by summing the hysteresis loops of the material in the fatigue process. As the number of load cycles increases, the area of the material hysteresis loop also gradually becomes larger. This tendency establishes a corresponding relationship to predict joint fatigue life. Shokrich and Lessard [17] established a fatigue cumulative damage analysis method based on a three-dimensional finite element model to analyze the fatigue life of laminate joints. This method starts by studying the fatigue performance of single-layer laminate and, based on the strength degradation, establishes the residual fatigue strength and residual stiffness degradation model of each single-layer laminate. In order to make the material fatigue degradation model applicable to arbitrary stress ratios, the method uses a regularized fatigue life prediction model [16]. Zhou [18] adopted equivalent stress to investigate the effect of multiaxial stress on the fatigue of composite bolted joints, and the joint stress distribution was calculated by finite element software (ABAQUS 6.14). Then, the equivalent stress of the joint was calculated by the Tsai-Hill criterion, and the accuracy of the equivalent stress method was analyzed by the S-N curve obtained from the comparison experiment.

The UMAT subroutine for the CFRP bolted joint fatigue was developed by Cai [19], and the theory of the micro-mechanics of failure (MMF) was extended to analyze the fatigue progressive failure and predict the strength of the bolted joint. The predicted strength accuracy of the MMF approach, as well as the classical Tsai-Wu and Hashin theories, was compared with the experimental results, which showed the MMF approach had the best accuracy. Ebadi-Rajoli [20] adopted a cohesive zone model to simulate the progressive damage of a composite bolted joint under mixed cyclic loading. Ge [21] studied the random fatigue life prediction method of metal. The frequency domain formula of the equivalent Lemaitre stress is adopted, and the corresponding method for estimating fatigue life under multiaxial random loading is developed based on the multiaxial S–N curve. The experiments and simulation results show that hydrostatic stress has a significant influence on multiaxial fatigue life.

Current research on the random fatigue of composite bolted joints is very limited. Through experiments and simulations, the random fatigue life of joints can be predicted, and the joint damage evolution and failure modes compared and analyzed. However, the method for the random fatigue life prediction of composite bolted joints can usually only calculate the fatigue life, and the failure mechanism cannot be studied by the existing method. Composite laminate consists of fiber and matrix, which has a complex failure mechanism. The failure modes are usually modeled by the finite element model, which can only be used by researchers in constant amplitude fatigue simulations. So, it is necessary to establish a random fatigue prediction method for composite bolted joints that can predict the random fatigue life and failure modes. The constant amplitude fatigue and random fatigue research methods are mixed in this paper, and the random fatigue life and failure modes can be predicted by the method established in our paper.

## 2. Experiments

The experiments consist of three parts: static strength experiments, constant amplitude fatigue experiments and random fatigue experiments. The amplitude of all fatigue experiments is selected based on the static strength of joints. As the stress ratio is R = −1, the tensile and compressive strengths need to be obtained through static experiments.

The composite laminates were made from T700 carbon fiber/YPH-26 epoxy resin provided by Guangwei Composite Materials Co., Ltd. (Weihai, China), and the bolt material was TC4 titanium alloy. The stacking sequence of the laminate was [45_3_/90_3_/−45_3_/0_3_]_s_. As shown in Figure 1, four kinds of joints were designed in the experiments to investigate the effect of the geometric parameters (GP), which were the distance (E) from the free edge to the center point of the hole, the width (W) of the joint and the diameter (D) of the hole. D was equal to 6 mm. Four kinds of joints are shown in Table 1, and the thickness of the laminate was 2.4 mm.

The experiments were carried out using the MTS 370 machine shown in Figure 2, which was manufactured by Mechanical Testing & Simulation Corporation (Eden Prairie, MN, USA). The static test tensile loading speed was 5 mm/min, and the compressive loading speed was 2 mm/min according to ASTM D5961 [22]. The displacement limit for loading was 4 mm, and the maximum applied load was 16 kN. To avoid the joint’s structure buckling in the early test stage, an anti-buckling device was designed. The fixture device is shown in Figure 3.

In the static and fatigue tests, 5 specimens were tested in every load condition. All the joint specimens were named for convenience in the analysis. The specimens were named in the following forms: A-B-C-D, A is the E/D, B is the W/D, C is the type of experiment, and D is the specimen number of the current test. For example, 2-3-ST-4 represents the joint with E = 12 mm and W = 18 mm and is the fourth specimen in the static tensile test. The abbreviations of experiment types are given as follows: ST is the static tensile test, SC is the static compressive test, CA stands for the constant amplitude fatigue test, and RF represents the random fatigue test.

### 2.1. Static Strength Experiments

In the static strength experiments, 4 kinds of joints were tested to obtain the tensile and compressive failure loads, and the comparison is given in Figure 4. The bars with a light color represent the tensile failure loads of joints, and the dark color bars represent the compressive failure loads of joints. With the increase in E/D, compared with the failure of the joint with E/D = 2, the tensile failure loads show small increases from 1.5% to 4.5%, which can be ignored. Compared with the failure of the joint with E/D = 2, the compressive failure loads show an obvious increase from 26.9% to 46.8%. With the increase in W/D, the failure loads of tension and compression all increase by 16.0% to 39.4%.

The common tensile failure modes of joints are net tension (N), shear out (S), bearing (B) and delamination (DE). In this experiment, some joints still buckled, even with the anti-buckling device. This is because the compressive failure load was close to the tensile failure load.

The failure modes of joints under tensile and compressive loads are listed in Table 2. T and C mean tensile and compressive loads. For the tensile test, the failure mode was net tension and delamination, and bearing occurred in the joints when the geometric parameter was 24. The increase in E/D had no effect on the joint’s failure mode, but the increase in W/D did have an effect on the failure mode. As shown in Figure 5, bearing failure happened at the edge of the joint hole. This is because the structure strength became greater when the geometric parameter W/D was increased, and the material compressive strength showed no change. Buckling appeared during the compressive experiments which can be seen in Figure 5 by the black arrows, and all the joints failure modes were delaminated.

### 2.2. Constant Amplitude Fatigue Experiments

A sinusoidal load was adopted in the constant amplitude fatigue experiments. The loading frequency was 10 Hz, and the stress ratio was R = −1. The loading amplitudes were selected by the percent of static strength. The power function was adopted to fit the S-N curves of the joints as shown below:
(1)$$S=AN^{b}$$

*S* is stress, *N* is the fatigue life of joint, *A* is the static strength, and *b* is the fitting parameter. The S-N curve is given in Figure 6.

The S-N curves show the tendency of the joint fatigue life with different geometric parameters. With the increase in E/D, the fatigue lives slightly increased and were raised within one magnitude. The increase in W/D made the fatigue life rise by one magnitude. The 2-4-CA and 4-3-CA joints showed a similar fatigue life, which means increasing the geometric parameters in different dimensions can raise the fatigue life.

The fitting parameters of the S-N curve formula are given in Table 3. Comparing the standard error, the dispersion of experimental data is related to the geometric parameters. The joint with bigger geometric parameters had a smaller dispersion, and it was influenced by the initial damage and the area between the layers.

The fatigue failure modes of the joints are shown in Table 4. All the joints had the failure modes of net tension and delamination. Delamination occurs easily in laminate, with obvious anisotropy caused by the ply sequence, which can be seen in our laminate with the ply sequence [45_3_/90_3_/−45_3_/0_3_]_s_. The joints with the W/D = 3 geometric parameter showed buckling failure. Joints with the W/D = 4 geometric parameter showed shear out and bearing failure. It is obvious that different failure modes happen to the joints when the W/D is changed, and an increase in W/D results in the bearing failure mode, which can increase the fatigue life significantly. As shown in Table 4, ◯ means the appearance of the failure mode.

### 2.3. Random Fatigue Experiments

The random fatigue load spectrum was generated by the Gaussian random process. The Gaussian random load spectrum had a stress ratio of R = −1, and it can be generated by the power spectral density (PSD). The function of the generated spectrum is shown in Equation (2).
(2)$$x\left(t\right)=\overset{+\infty }{\underset{-\infty }{\int }}e^{i\omega t}\sqrt{h\left(\omega \right)}e^{-2\pi Rj}d\omega$$

The PSD of the spectrum was 0.75 kN^2^/Hz, and the frequency was 1~4 Hz, and these parameters were used to generate a 120 s spectrum. The random spectrum generated by MATLAB 2016 is shown in Figure 7.

The random spectrum was counted to obtain the amplitude probability distribution. The amplitude probability distribution obeys Weibull distribution. The formula of Weibull distribution is shown in Equation (3).
(3)$$f\left(x,\lambda ,k\right)=\left\{\begin{array}{c} \frac{k}{\lambda }{\left(\frac{x}{\lambda }\right)}^{k-1}e^{-{\left(x/\lambda \right)}^{k}}\;x\geq 0\\ \;\;\;\;0\;\;\;x<0 \end{array}\right.$$

As shown in Figure 8, *x* is the random variable, *λ* is the scale parameter, and *k* is the shape parameter. Drawing the amplitude distribution of the 120 s spectrum and 5000 s spectrum, we found the Weibull distribution was similar, so the 120 s spectrum could represent the true random time history. Different amplitudes were applied to the joints in the constant amplitude experiments, so the load level was different in the random fatigue experiments. On the four kinds of joints, 104%, 127% and 155% amplitude of the origin spectrum were applied.

The failure mode of joints under random fatigue loading is shown in Figure 9. All the joint failure modes were net tension, delamination and bearing. The random loading spectrum was variable amplitude loading, so when the joints were under a bigger load amplitude, the fiber and matrix damage modes were similar to the failure modes under the static tensile load.

In the random loading experiments, the joint failure modes were net tension and bearing. For the same joints, buckling happened in the constant amplitude experiments. This is because the loading amplitude caused a different damage mode, and the composite matrix damage grew slowly under constant amplitude loading compared to the random load. The failure mode of the random loading joints was similar to the static experiment, which can be seen in Table 5, and ◯ means the appearance of the failure mode. The matrix cracking grew quickly when the loading amplitude was bigger than the structure-limited load.

## 3. Failure Mode and Life Prediction Method

Based on the linear cumulative damage hypothesis, the fatigue life of the joint was established as a function of the loading amplitude, and then the random life prediction was carried out by the amplitude distribution of random loading. This kind of method needs testing for structure fatigue. Another method is the equivalent load method, which is used to obtain the equivalent constant amplitude fatigue loading of the random loading spectrum, and this kind of method only needs to obtain the material’s properties compared with the former one. Finite element modeling was set up based on the material performance degradation model, and the equivalent random life and damage to the joints were calculated.

### 3.1. Random Fatigue Life Prediction

The prediction of random life is based on the Miner linear cumulative damage criteria and amplitude Weibull distribution. In Miner linear damage criteria, the damage factor is related to the loading amplitude, and the linear criteria ignore the effect of load sequence, so the total damage of current loading is:(4)$$D=\sum_{i=1}^{n}\frac{1}{N_{i}}$$

The damage factor can be obtained from the S-N fitting curve. The formula of the S-N fitting curve is Equation (1). The damage factor is:(5)$$\frac{1}{N}={\left(\frac{S}{A}\right)}^{1/b}$$

With different amplitudes, different damage factors can be calculated. For the random load spectrum, load amplitudes obey Weibull distribution so multiply the damage factor and the probability density function. Then, the integral of this function on the amplitude field and the damage factor of random fatigue are obtained.
(6)$$\frac{1}{N_{R}}=\int_{A\min}^{A\max}{\left(\frac{S}{A}\right)}^{1/b}\frac{k}{\lambda }{\left(\frac{S}{\lambda }\right)}^{k-1}e^{-\left(x/\lambda \right)k}dS$$
where $1/N_{R}$ is the expectation damage factor of random fatigue, and $N_{R}$ is the fatigue life of random fatigue. This formula can predict the random fatigue life using the Weibull parameters and the S-N curve fitting parameters. The predicted life is shown in Table 6.

The predicted life is similar to the experimental life data, and the calculation errors show that the result is in an acceptable range.

### 3.2. Equivalent Fatigue Life Prediction

The finite element calculation of the random fatigue process can predict the fatigue life of the joint, but the calculation time required for direct loading is longer. The equivalent load was used to equivalently process the random loading, and then the finite element model was loaded with the constant amplitude. The eight-node brick elements (C3D8R) were used in this model, as shown in Figure 10. Mesh was refined around the hole due to the stress concentration around the hole, as shown in Figure 10. The coefficient of friction was 0.2. With cyclic loading, random fatigue calculation results can be obtained quickly.

By predicting the fatigue life of joints using finite element software, the joint damage process and failure life can be obtained accurately. The UMAT subroutine of ABAQUS 6.14 FEM software can calculate the damage evolution process through the material properties’ degradation. Under random fatigue conditions, it takes a long time for the software to find the fatigue life in the upper time domain. If we can find the constant fatigue loading equivalent to the random fatigue loading, this will greatly reduce the calculation.

A modified Hashin criterion was adopted for the fatigue failure criteria.
1.Fiber tensile fatigue failure (FT), $\sigma_{1}>0$: (7)$${\left(\frac{\sigma_{1}}{X_{T}(n,\sigma ,R)}\right)}^{2}+{\left(\frac{\sigma_{12}}{S_{12}(n,\sigma ,R)}\right)}^{2}+{\left(\frac{\sigma_{13}}{S_{13}(n,\sigma ,R)}\right)}^{2}\geq 1$$2.Fiber compressive fatigue failure, $\sigma_{1}<0$:(8)$${\left(\frac{\sigma_{1}}{X_{C}\left(n,\sigma ,R\right)}\right)}^{2}\geq 1$$3.Matrix tensile fatigue failure, $\sigma_{2}>0$:(9)$${\left(\frac{\sigma_{2}}{Y_{T}(n,\sigma ,R)}\right)}^{2}+{\left(\frac{\sigma_{12}}{S_{12}(n,\sigma ,R)}\right)}^{2}+{\left(\frac{\sigma_{23}}{S_{23}(n,\sigma ,R)}\right)}^{2}\geq 1$$4.Matrix compressive fatigue failure, $\sigma_{2}<0$:(10)$${\left(\frac{\sigma_{2}}{Y_{C}(n,\sigma ,R)}\right)}^{2}+{\left(\frac{\sigma_{12}}{S_{12}(n,\sigma ,R)}\right)}^{2}+{\left(\frac{\sigma_{23}}{S_{23}(n,\sigma ,R)}\right)}^{2}\geq 1$$5.Fiber matrix shear fatigue failure, $\sigma_{1}<0$:(11)$${\left(\frac{\sigma_{1}}{X_{C}(n,\sigma ,R)}\right)}^{2}+{\left(\frac{\sigma_{12}}{S_{12}(n,\sigma ,R)}\right)}^{2}+{\left(\frac{\sigma_{13}}{S_{12}(n,\sigma ,R)}\right)}^{2}\geq 1$$6.Normal tensile fatigue failure (delamination), $\sigma_{3}>0$:(12)$${\left(\frac{\sigma_{3}}{Z_{T}(n,\sigma ,R)}\right)}^{2}+{\left(\frac{\sigma_{13}}{S_{13}(n,\sigma ,R)}\right)}^{2}+{\left(\frac{\sigma_{23}}{S_{23}(n,\sigma ,R)}\right)}^{2}\geq 1$$7.Normal compressive fatigue failure (delamination), $\sigma_{3}<0$:(13)$${\left(\frac{\sigma_{3}}{Z_{C}(n,\sigma ,R)}\right)}^{2}+{\left(\frac{\sigma_{13}}{S_{13}(n,\sigma ,R)}\right)}^{2}+{\left(\frac{\sigma_{23}}{S_{23}(n,\sigma ,R)}\right)}^{2}\geq 1$$

The degradation of material properties, including its strengths and modulus, is as follows:(14)$$\frac{S\left(n,\sigma ,R\right)-\sigma }{\left(S_{s}-\sigma \right)}={\left[1-{\left(\frac{\log\left(n\right)-\log\left(.25\right)}{\log\left(N_{f}\right)-\log\left(.25\right)}\right)}^{\beta }\right]}^{\frac{1}{\alpha }}$$
(15)$$\frac{E\left(n,\sigma ,R\right)-\frac{\sigma }{\varepsilon_{f}}}{E_{s}-\frac{\sigma }{\varepsilon_{f}}}={\left[1-{\left(\frac{\log\left(n\right)-\log\left(.25\right)}{\log\left(N_{f}\right)-\log\left(.25\right)}\right)}^{\beta }\right]}^{\frac{1}{\alpha }}$$

As shown in Equations (12) and (13), $S\left(n,\sigma ,R\right)$ stands for the residual strength; $S_{s}$ is the static strength; $E\left(n,\sigma ,R\right)$ is the residual stiffness; $E_{s}$ is the static stiffness; $\varepsilon_{f}$ is the average strain to failure; n respects the number of applied cycles; σ is the magnitude of applied maximum stress; $N_{f}$ is the fatigue life at σ; R is the stress ratio; and α and β are the experimental curve fitting parameters.

When the composites are under fatigue loading, Poisson’s ratios $\nu_{12}=\nu_{13}=0.3$ and $\nu_{23}=0.42$ do not change during the numerical calculation in the subroutine. The other properties degradations follow the Table 7 rules. In Table 8, all the experimental curve fitting parameters (progressive failure), static properties and the final failure degradation (sudden failure) caused by the failure criteria or fatigue life are listed in Table 9.

Assuming the stress–strain characteristics and the structural response are all in a steady state under random loading, the effect of the random loading sequence and interactions between the loading is ignored for the mean life.

From an energy point of view, the hysteresis energy absorbed by each cycle of the material is Δ*W*. When the total absorbed energy *W* reaches a critical value, the joint fatigue failure occurs. The energy absorbed by the material stress amplitude cycle is the stable hysteresis loop area, which is approximated by the Morrow formula:(16)$$\Delta W=3{\left(S_{a}^{n+1}/K\right)}^{1/n}$$

In the formula, *n* is the material cycle strain hardening parameter. For narrow-band random stress waves, the cumulative energy absorbed per unit time is:(17)$$W=\int_{0}^{\infty }\Delta W\left(S\right)dn\left(S\right)$$

*dn*(*S*) is the number of stress cycles between the stress amplitudes *S* and *S* + *dS* per unit of time. In narrowband conditions:(18)$$dn\left(S\right)=P\left(S\right)f_{0}dS$$

According to the random signal theory, the stress cycle frequency per unit time is:(19)$$f_{0}=\sqrt{\int f^{2}S\left(f\right)df/\int S\left(f\right)df}$$

The narrow band stress cycle probability density is:(20)$$P\left(S\right)=\left(S/\sigma^{2}\right)e^{-S^{2}/2\sigma^{2}}$$

$\sigma^{2}$ is the mean square of the random stress wave. Substituting Equations (16), (18) and (20) into Equation (17) gives:(21)$$W=\int_{0}^{\infty }\frac{3S^{\left(n+1\right)/n}}{K^{1/n}}f_{0}\frac{S}{\sigma^{2}}e^{-S^{2}/2\sigma^{2}}dS=\frac{3f_{0}2^{\left(1+n\right)/2n}\Gamma \left[\left(1+3n/2n\right)\right]}{K^{1/n}}\sigma^{\left(1+n\right)/n}$$

*K* is the material cycle strength coefficient. Γ(*x*) is the gamma function.

For the narrowband random stress, the effect of broadband energy on narrowband energy is reflected by the correction factor *λ* [23]. In this experiment, the loading spectrum is a narrow band, and the frequency shows that the effect of broadband can be ignored, so *λ* is considered to be 1.

The condition of the load spectrum conversion is that the damage of the random spectrum and the constant amplitude input is equal in a given time; that is, the cumulative energy of the two is consistent, and this is the concept of equal energy damage. In a steady state, the equivalent conditions of the random spectrum and the constant spectrum can be expressed as:(22)ΔWFt=Wt

*F* is the frequency of constant amplitude loading. Substituting the (16) and (21) formulas into the (22) formula gives:(23)$$S_{e}=\sqrt{2}{\left[\frac{f_{0}\lambda }{F}\Gamma \left(\frac{1+3n}{2n}\right)\right]}^{n/\left(n+1\right)}\sigma$$

*S_e_* is the equivalent constant loading spectrum, and the magnitude of the equivalent constant amplitude loading spectrum can be obtained by the material cycle strain hardening parameter *n* and the RMS of the random loading spectrum. For composites, *n* is approximately equal to 1. In the static test, the steady-state phase of the load-displacement curve was almost linear, as shown in Figure 11. The black dash line is a linear line, so the *n*-value of the joint structure is also approximately 1. The RMS of the random spectrum is 1.5.

For 2-3-RF joints, the equivalent constant amplitude was 4.8 kN. The load was applied to the FEA model, and the fatigue life and damage state at the amplitude were calculated, which can be seen in Table 10.

Comparing the failure mode of the joint, it can be seen that fiber damage has little effect on the joint failure, and the joint failure is mainly due to the cracking and delamination of the matrix. The simulation calculation also shows that the predicted joint cracking position is consistent with the actual situation, which indicates that the method is correct for the prediction of random fatigue life. As shown in Figure 12, the red colour is total failure, the blue colour is no damage, and the other colours mean not total failure but damage.

The long dashed line in Figure 13 indicates that the test life and predicted life error are within two times, and the solid line indicates that the error is within three times. It can be seen that all prediction errors are within two times, the prediction of the cumulative damage method is slightly larger, and the FEA prediction is conservative. The FEA calculation results can also reflect the damage to the joint, and the reliability of the fatigue life prediction can be further verified by comparing the actual damage to the joint.

## 4. Discussion of Two Prediction Methods

Both the finite element model established by material degradation and the linear cumulative damage model can predict the fatigue life of joints. The long dashed line in Figure 13 indicates that the error between the test and the fatigue life is within two times, and the long solid line in the figure indicates that the error between the test and the fatigue life is within three times. The prediction results of both methods show good accuracy, and all prediction results are included in a scatter band of the factor of 2. Figure 13Random fatigue life comparison.
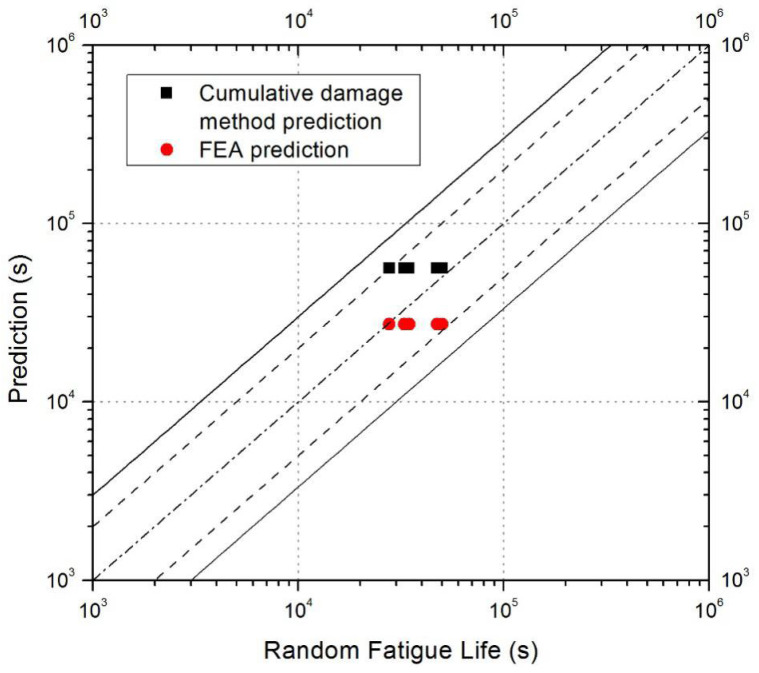


The linear cumulative damage model assumes that the joint damage and the load amplitude are in a power function relationship. The relationship between the fatigue life of the joint and the load amplitude is obtained by the normal amplitude fatigue experiment, and the random fatigue can be obtained by the amplitude distribution function of the random loading life. This method can predict the random fatigue of joints with fewer experimental amounts and a simpler loading method.

The finite element model prediction method needs to establish a material subroutine through material property degradation law to achieve fatigue calculation. This method requires a large number of material basic experiments at the beginning to obtain the basic mechanical properties and degradation laws of the materials. From Figure 14, the test results are similar in constant and random fatigue loading conditions, which shows the equivalent fatigue life prediction. Although it takes longer time than the cumulative damage calculation method, when the constitutive model of the material is established, the method can be used to predict the fatigue problem of different structures and has good generalization.

Comparing the two prediction methods, both methods give good life prediction results. The difference between the two methods is that the FEA prediction results are relatively conservative. For different application situations, the experimental or computational method can be used to effectively predict the random life of joints.

## 5. Conclusions

In this paper, the static and fatigue properties of composite bolted joints were investigated by experimental and numerical methods.

In the static test, with the increase in E/D and W/D, the failure loads show an obvious increase. In the constant amplitude fatigue experiments, the failure modes are similar to the static experimental results, and the critical failure mode is net tension.

The novel prediction method of random fatigue life is established in this paper. The Miner linear cumulative damage method and equivalent loading method are compared with each other, and the predicted random fatigue life with these two methods are within a scatter band of the factor of 2, which indicates that both methods can effectively predict the random fatigue life.

## Figures and Tables

**Figure 1 materials-17-02740-f001:**
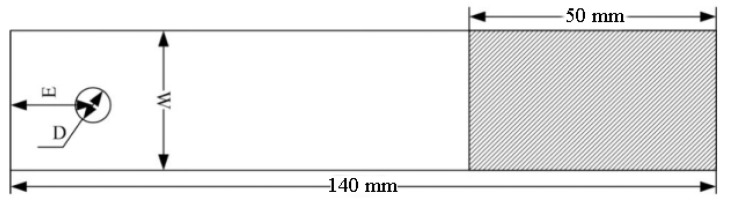
Joint geometric parameters.

**Figure 2 materials-17-02740-f002:**
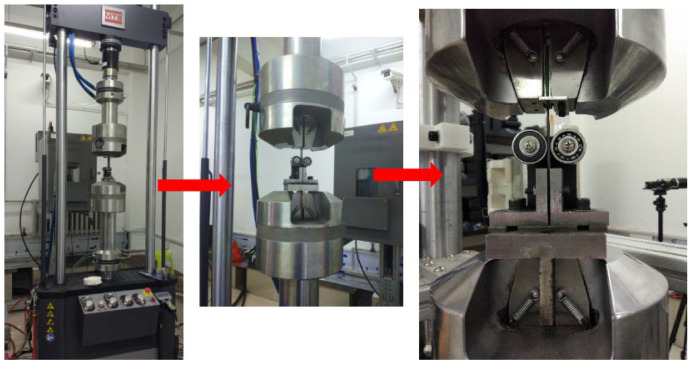
Experimental equipment.

**Figure 3 materials-17-02740-f003:**
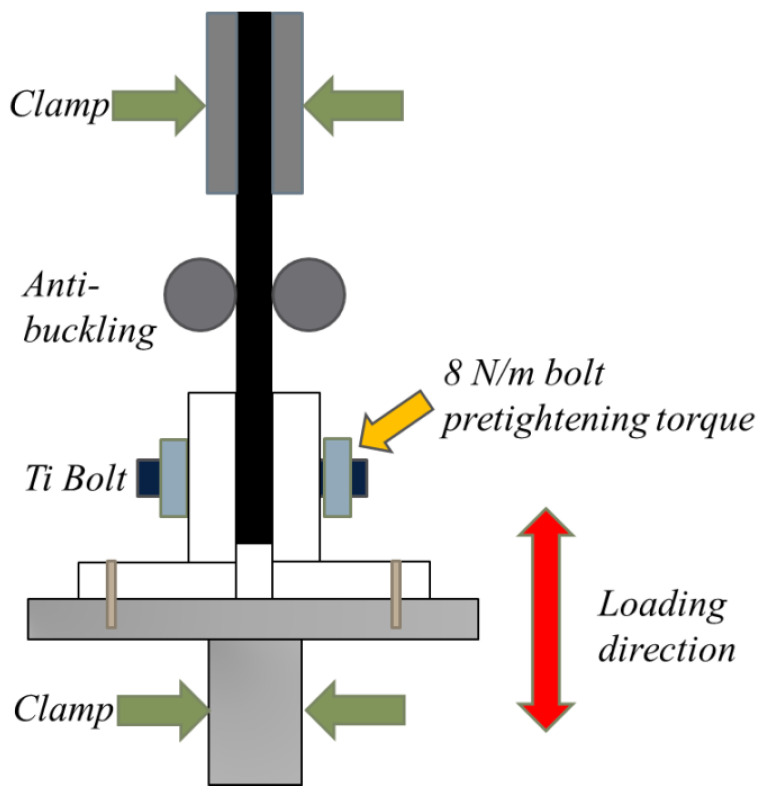
Loading device.

**Figure 4 materials-17-02740-f004:**
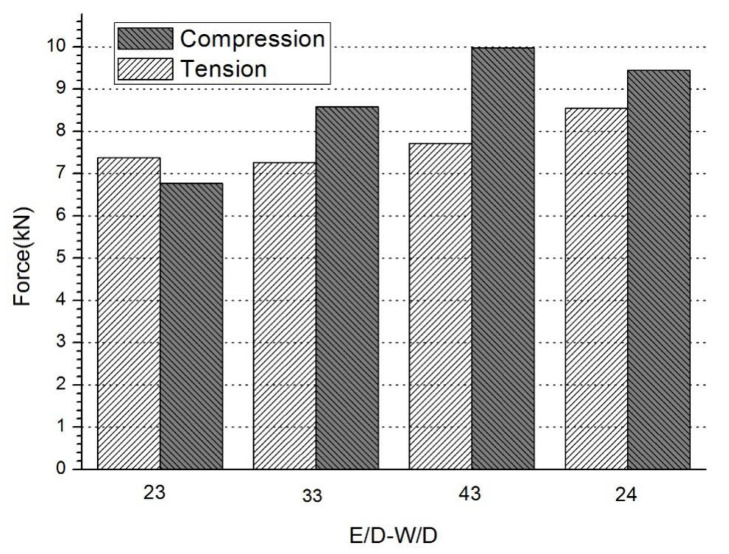
Strength of all the joints.

**Figure 5 materials-17-02740-f005:**
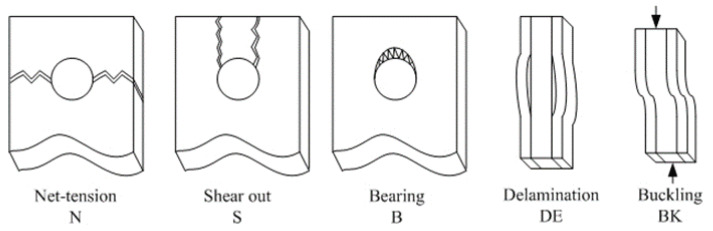
Failure modes of composite bolted joints.

**Figure 6 materials-17-02740-f006:**
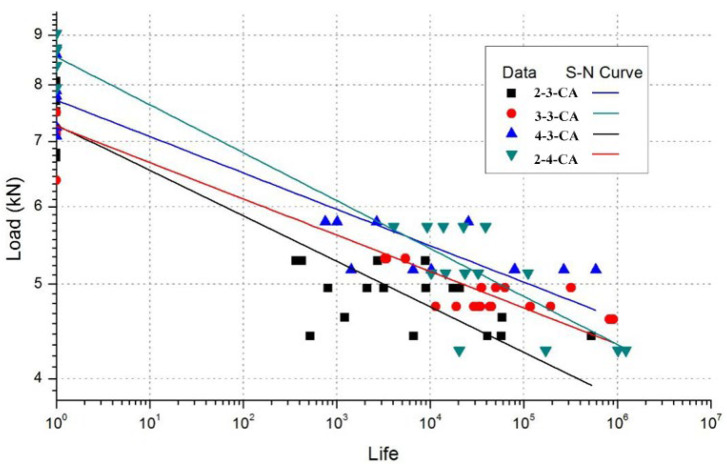
S-N curves of joints.

**Figure 7 materials-17-02740-f007:**
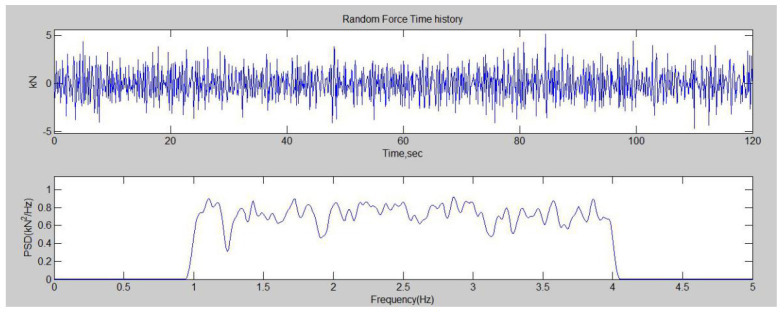
Random loading spectrum and PSD.

**Figure 8 materials-17-02740-f008:**
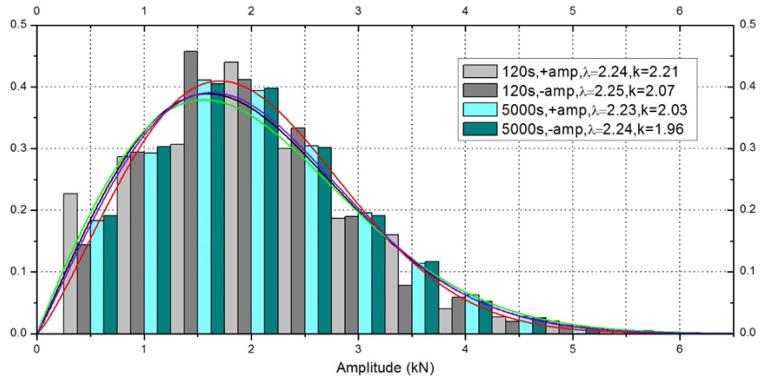
Amplitude distribution of random spectrum.

**Figure 9 materials-17-02740-f009:**
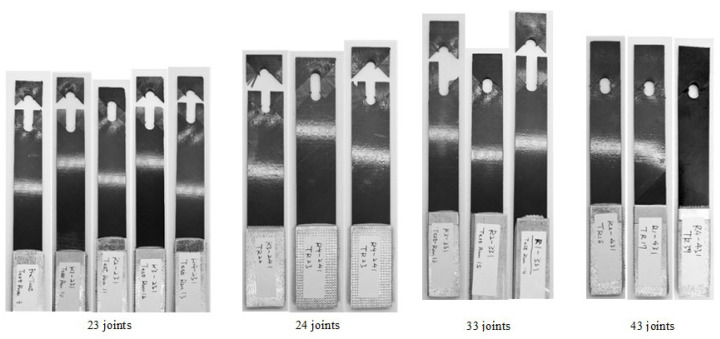
Failure modes of joints under random fatigue loading.

**Figure 10 materials-17-02740-f010:**
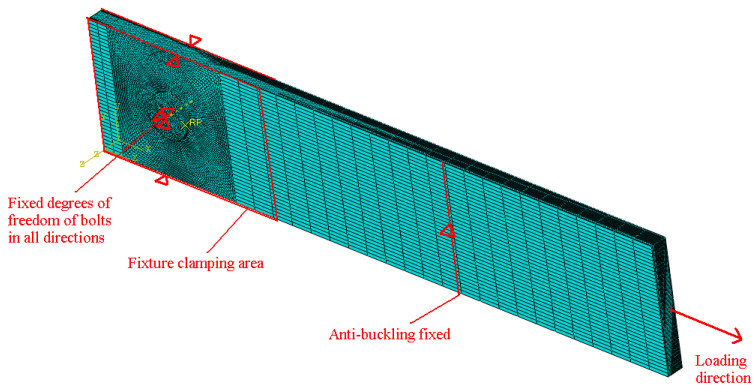
Finite element model.

**Figure 11 materials-17-02740-f011:**
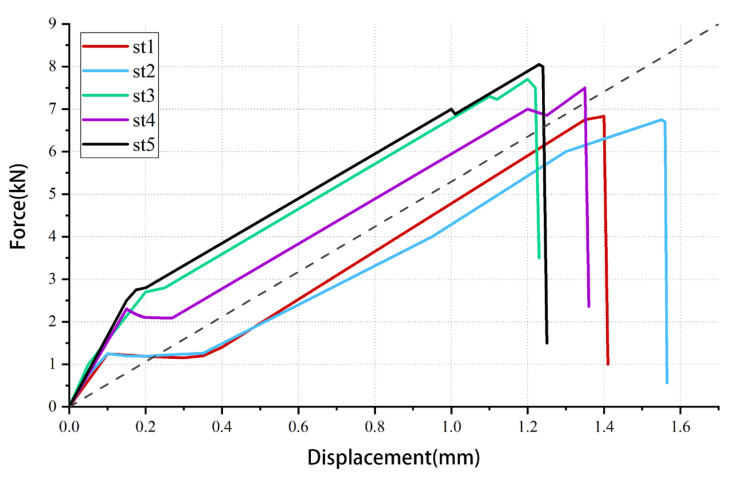
The curve can be regarded as linear when statically stable loading.

**Figure 12 materials-17-02740-f012:**
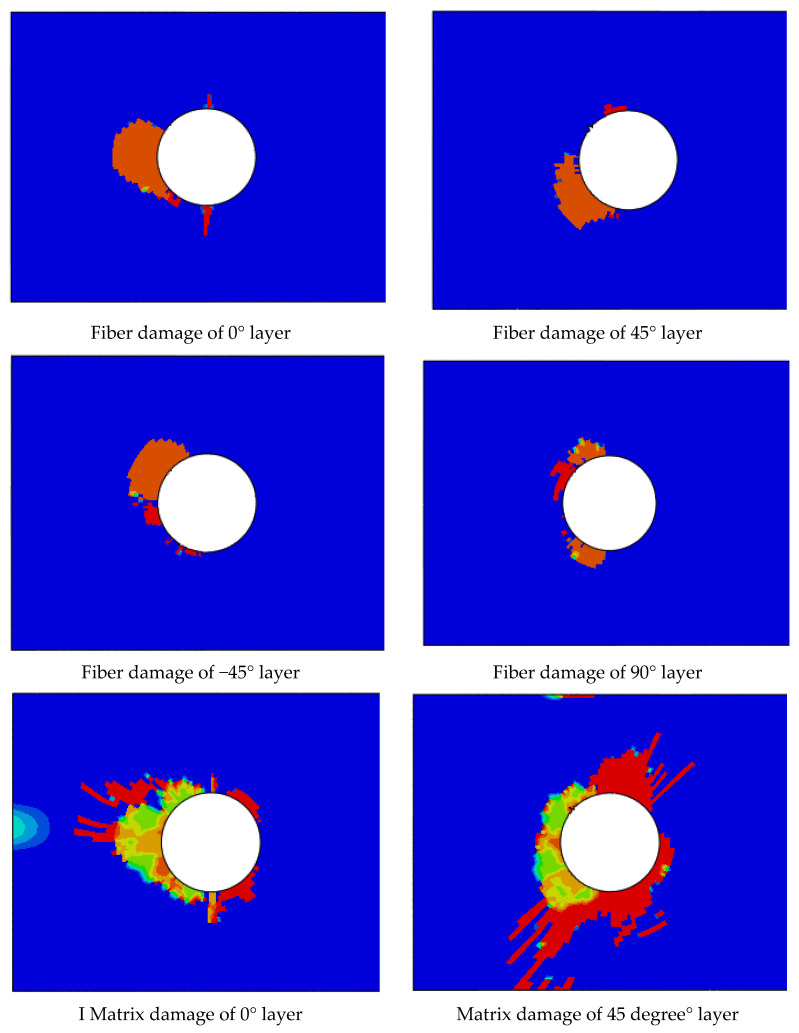
Damage simulation of joint.

**Figure 14 materials-17-02740-f014:**
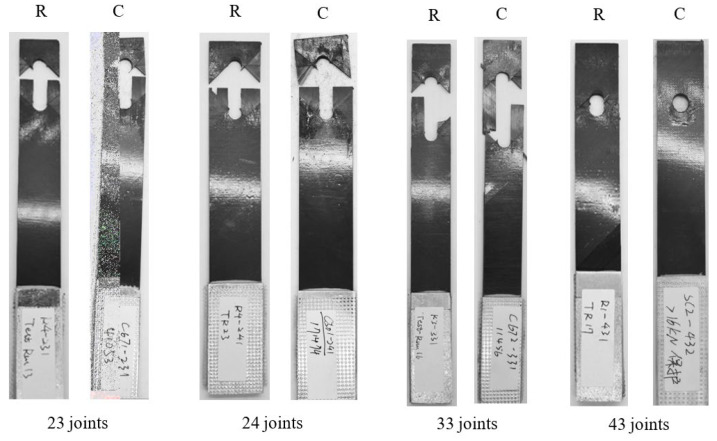
Test result comparison of constant and random fatigue loading.

**Table 1 materials-17-02740-t001:** Four kinds of joints with different geometrics.

	No. 1	No. 2	No. 3	No. 4
E/D	2	3	4	2
W/D	3	3	3	4

**Table 2 materials-17-02740-t002:** Static failure modes of joints.

	Failure Modes	N		S		B		DE		BK	
Geometric Parameter Ratios											
23		T						T	C		C
33		T						T	C		C
43		T						T	C		C
24		T				T	C	T	C		C

**Table 3 materials-17-02740-t003:** Fitting parameters of S-N curves.

	Fitting Parameters	A	b	*Standard Error*
Geometric Parameters Ratios				
2-3-CA		7.28	−0.04537	0.00256
3-3-CA		7.26	−0.03725	0.00221
4-3-CA		7.72	−0.03599	0.00245
2-4-CA		8.55	−0.04912	0.00203

**Table 4 materials-17-02740-t004:** Fatigue failure modes of joints.

	Failure Modes	N	S	B	DE	BK
Geometric Parameters Ratios						
2-3-CA		◯			◯	◯
3-3-CA		◯			◯	◯
4-3-CA		◯			◯	◯
2-4-CA		◯	◯	◯	◯	

**Table 5 materials-17-02740-t005:** Failure mode comparison.

	N	S	B	DE	BK
Static Experiment	◯		◯	◯	◯
CA	◯	◯	◯	◯	◯
RF	◯		◯	◯	

**Table 6 materials-17-02740-t006:** Random fatigue life comparison.

GP	Load Level	Average Life (s)	Prediction (s)	Error
2-3-RF	104%	38,573.20	56,075.56	45.4%
2-4-RF	155%	45,379.00	29,009.72	36.1%
3-3-RF	127%	21,503.33	13,169.73	38.8%
4-3-RF	155%	38,855.33	30,945.35	21.4%

**Table 7 materials-17-02740-t007:** T700 property parameters.

Properties	E_11_	E_22_	E_33_	G_12_	G_13_	G_23_	S_23_
Static magnitude	147.0 GPa	9.0 GPa	9.0 GPa	5.0 GPa	5.0 GPa	3.0 GPa	42 MPa
X_T_	X_C_	Y_T_	Y_C_	Z_T_	Z_C_	S_12_	S_13_
2004 MPa	1197 MPa	53 MPa	204 MPa	53 MPa	204 MPa	137 MPa	137 MPa

**Table 8 materials-17-02740-t008:** T700 fatigue parameters.

Fatigue Parameters		
	A	β
E_11_	14.5700	0.3024
E_22_	14.7700	0.1155
E_33_	14.7700	0.1155
G_12_	0.7000	11.0000
G_13_	0.7000	11.0000
G_23_	0.7000	11.0000
X_T_	10.0300	0.4730
X_C_	49.0600	0.0250
Y_T_	9.6280	0.1255
Y_C_	67.3600	0.0011
Z_T_	9.6280	0.1255
Z_C_	67.3600	0.0011
S_12_	0.1600	9.1100
S_13_	0.1600	9.1100
S_23_	0.1600	9.1100

**Table 9 materials-17-02740-t009:** T700 failure modes.

Failure Modes							
	FT	FC	F-M S	MT	MC	NT	NC
E_11_	0	0					
E_22_	0	0		0	0		
E_33_	0	0				0	0
G_12_	0	0	0				
G_13_	0	0					
G_23_	0	0					
X_T_	0	0					
X_C_	0	0					
Y_T_	0	0		0			
Y_C_	0	0			0		
Z_T_	0	0				0	
Z_C_	0	0					0
S_12_	0	0	0				
S_13_	0	0					
S_23_	0	0					

**Table 10 materials-17-02740-t010:** Random fatigue life comparison with two methods.

Test Fatigue Life (s)	Cumulative Damage Method Prediction (s)	FEA Method Prediction (s)
38,573.20	56,075.56	27,239.18

## Data Availability

The original contributions presented in the study are included in the article, further inquiries can be directed to the corresponding author/s.

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
