# Peer review of "Fatigue Analysis of Composite Bolted Joints under Random and Constant Amplitude Fatigue Loadings"

_materials, 2024, doi:10.3390/ma17112740_

Round 1
Reviewer 1 Report
Comments and Suggestions for Authors
After carefully reviewing this study on random fatigue analysis aimed at determining fatigue life under various amplitude loadings, I believe it is suitable for publication in the Materials journal. However, there are some comments to enhance the quality of this work:
-
In the abstract, it would be beneficial to include quantitative results rather than solely relying on qualitative comparisons.
-
For the keywords, incorporating more specific terms could assist readers in locating your article, not limited to "joint for composite."
-
The rationale for utilizing random fatigue is not sufficiently explained. Why did the authors choose this method over design of experiments and pre-testing followed by normal fatigue?
-
The final paragraph of the introduction lacks clarity. It should outline the novelty and objectives of the study, which are currently difficult to discern.
-
Section 2 should commence with an introduction to your raw materials or base material before delving into the fatigue test description.
-
Regarding "R," are you certain it's not 0.1?
-
Additional properties of carbon fiber specimens should be provided. In Figure 1, it would be advantageous to include thickness and other dimensions.
-
When conducting tensile or fatigue tests, referencing relevant standards and protocols is advisable.
-
All equipment utilized in the study should be specified with model and manufacturer details.
-
Placing all diagrams and plots in enclosed boxes would enhance clarity.
-
In Tables 3 and 6, clarification is needed regarding the standard error. Is it referring to standard deviation?
-
What does the y-axis represent in Figure 8?
-
Further elaboration is required on the modeling and simulation conducted with Abaqus, particularly regarding sample meshing.
-
Consistency in decimal usage is recommended across all tables, especially in Table 8.
-
Why is the damage in the fiber damage of the 90-degree layer less than that of the 45-degree layer?
-
All plots and diagrams should be enclosed, akin to Figure 13.
-
In Figure 13, the meanings of the dashed lines should be clearly labeled.
-
Based on the multitude of results, restructuring the conclusion into separate bullet points, each addressing a main result with supporting evidence, would improve clarity.
-
Incorporating more recent references, particularly from 2023-2024, is necessary to bolster the study's relevance.
Reviewer 2 Report
Comments and Suggestions for Authors
This paper examines random fatigue life and failure modes of joints via two calculation methods, based on static, constant amplitude fatigue, and random fatigue tests. It predicts random fatigue life using the linear cumulative damage and equivalent loading finite element methods, showing good agreement with experimental results.
The topic is interesting and relevant. The paper is well written and organized. There are some points that would need the authors' attention before considering the work for publication in Materials.
The authors need to clarify the application of the different types of joints analyzed.
In figure 6 don't include the static results. The S-N line should be fitted to the data only suing the fatigue results. The necessary corrections should be also made in the text, accordingly.
The introduction can be enhanced considering the most recent works published in this filed such as: https://doi.org/10.1016/j.jajp.2022.100098 and https://doi.org/10.1016/j.mechmat.2020.103322
Table 1 should also include the values of each parameter for each configuration in addition to the ratio presented. Also this table should include the code assigned to each configuration to make it easier for the readers to follow the text and figures presented in the following sections.
Considering the data given in Table 4 and Figure 6 the authors should further discuss the relation between the failure mechanism and the fatigue life.
Enhance the clarity of Figure 11 by improving its quality and utilizing distinct patterns for each curve to facilitate better comprehension.
Comments on the Quality of English LanguageOverall, the English is fine but minor corrections can be made to enhance the paper in terms of English.
Round 2
Reviewer 1 Report
Comments and Suggestions for Authors
No more comments.